# Matrix vesicles promote bone repair after a femoral bone defect in mice

**Yuya Mizukami[1], Naoyuki Kawao[1], Yoshimasa Takafuji[1], Takashi Ohira[1], Kiyotaka Okada[1], Jun-Ichiro Jo[2,3], Yasuhiko Tabata[2], Hiroshi Kaji[1]***

1 Department of Physiology and Regenerative Medicine, Kindai University Faculty of Medicine, Osakasayama, Osaka, Japan, 2 Laboratory of Biomaterials, Department of Regeneration Science and Engineering, Institute for Life and Medical Sciences, Kyoto University, Kyoto, Japan, 3 Department of Biomaterials, Osaka Dental University, Hirakata, Osaka, Japan

* hkaji@med.kindai.ac.jp

## Abstract

Matrix vesicles (MtVs) are one of the extracellular vesicles (EVs) secreted by osteoblasts. Although MtVs have a classically-defined function as an initiator of ossification and recent findings suggest a role for MtVs in the regulation of bone cell biology, the effects of MtVs on bone repair remain unclear. In the present study, we employed collagenase-released EVs (CREVs) containing abundant MtVs from mouse osteoblasts. CREVs were administered locally in gelatin hydrogels to damaged sites after a femoral bone defect in mice. CREVs exhibited the characteristics of MtVs with a diameter <200 nm. The local administration of CREVs significantly promoted the formation of new bone with increases in the number of alkaline phosphatase (ALP)-positive cells and cartilage formation at the damaged site after the femoral bone defect. However, the addition of CREVs to the medium did not promote the osteogenic differentiation of ST2 cells or the ALP activity or mineralization of mouse osteoblasts *in vitro*. In conclusion, we herein showed for the first time that MtVs enhanced bone repair after a femoral bone defect partly through osteogenesis and chondrogenesis in mice. Therefore, MtVs have potential as a tool for bone regeneration.

## Introduction

Bone tissues possess a high regeneration capacity, and minor bone defects are generally repaired without the need for clinical interventions [1, 2]. Various bone cells, such as macrophages, mesenchymal stem cells, chondrocytes, osteoblasts, and osteoclasts, cooperatively participate in the bone repair process after bone injury during bone regeneration [3, 4]. Numerous specific molecular signaling pathways and activated cellular networks have been elucidated in bone regeneration [5]. However, high-quality clinical interventions are needed for bone regeneration under a number of pathological conditions. Although autologous bone grafts are currently the standard treatment for critical-sized bone defects [6], the requirement of second operation and the acquisition of appropriate graft tissues are limitations of autologous bone grafts [6]. To resolve these issues, the further clarification of bone regeneration mechanisms and the development of new treatment methods are needed.

**Data Availability Statement:** All relevant data are within the paper and its Supporting Information files.

**Funding:** The study was supported by the following grants: the 2021 Kindai University Research

Enhancement Grant (No. SR06) to Y. M. (https://
www.kindai.ac.jp/english/); a JSPS KAKENHI
Grant-in-Aid for Early Career Scientists (No.
22K16755) to Y. M and Grants-in-Aid for Scientific
Research (No. C:20K09514) to H.K. (https://www.
jsps.go.jp/english/); Takeda Science Foundation to
Y.T. (https://www.takeda-sci.or.jp/en/); The Salt
Science Research Foundation (No. 22C1) to H.K.
(https://www.saltscience.or.jp/english.html). The
funders had no role in study design, data collection
and analysis, decision to publish, or preparation of
the manuscript.

**Competing interests:** The authors have declared
that no competing interests exist.

Extracellular vesicles (EVs) are lipid membrane-enclosed microstructures with diameters of 30–1000 nm that are secreted by most cells. EVs comprise lipid bilayers, membrane proteins, and intravesicular cargo, which includes various proteins and nuclear acids, and they possess the capacity to exchange cargo between cells or organs [7, 8]. EVs act as signaling vehicles in normal cell homeostatic processes or pathological states [9–12]. A recent study showed using *in vivo* imaging that EVs secreted from osteoblasts were taken up by other osteoblasts in the calvaria [13]. Deng *et al*. reported that osteoblast-derived EVs promoted osteoclast formation through receptor activator of nuclear factor κB ligand (RANKL) expressed on EVs [14]. Moreover, Uenaka *et al*. showed that miR-143 in osteoblast-derived EVs increased RANKL expression in osteoblasts, thereby promoting bone resorption *in vivo* [13]. These findings suggest that osteoblast-derived EVs regulate bone remodeling [15]. On the other hand, EVs released by mineralization-induced MC3T3-E1 cells, a mouse preosteoblast cell line, promoted the osteoblast differentiation of bone marrow-derived mesenchymal stem cells with an improved capacity for ovariectomy-induced osteoporosis in mice [16], indicating the potential of osteoblast-derived EVs as a tool for bone therapy.

During ossification, mature osteoblasts release EVs into the extracellular matrix, which are called "matrix vesicles" (MtVs) [17]. MtVs have a diameter of 100–300 nm and bind to collagen [18, 19]. MtVs exhibit high tissue non-specific alkaline phosphatase (ALP) activity that hydrolyzes extracellular inorganic pyrophosphate, an inhibitor of hydroxyapatite formation [20, 21]. Therefore, the initiation of hydroxyapatite formation is a classically established function of MtVs. However, recent evidence suggests that MtVs mediate cell signaling and cell-cell communication, similar to other types of EVs, because MtVs possess bone morphogenetic proteins and microRNAs [22, 23]. Minamizaki *et al*. reported that osteoblast-derived EVs collected from a collagenase-treated osteoblast culture, which are rich in MtVs, inhibited osteoclast formation via the transfer of miR-125b [24], suggesting a role for MtVs in the regulation of bone remodeling. Since MtVs in the bone matrix are released after bone injury, we speculated that MtVs, which are abundant under bone regenerative conditions, regulate bone repair after bone injury.

Collagenase-released EVs (CREVs) are collected by the collagenase digestion of a mouse osteoblast culture and then subjected to the sequential centrifugation method. CREVs have been used in a number of studies as putative MtVs [24, 25]. Cationized gelatin hydrogels are suitable for the local sustained release of negatively charged molecules due to degradation [26]. A femoral bone defect model has generally been used to investigate the therapeutic effects of substitute biomaterials on bone repair and the bone repair process in mice as a representative model of the bone repair process after bone defects in various bones as well as a femoral bone defect [27, 28]. On the other hand, a calvarial bone defect model is sometimes useful for assessing bone repair through membranous ossification. Since the repair of the calvaria after a bone defect is difficult without the addition of strong osteogenic factors, such as bone morphogenetic proteins, we selected the femoral bone defect model to investigate the effects of osteoblast-derived CREVs on bone repair in mice.

Therefore, the present study investigated the effects of locally administered CREVs, putative MtVs, on bone repair after a femoral bone defect in mice.

## Materials and methods

### Materials

Anti-β-actin (#4970) and anti-PDGFRα (#3174S) antibodies were purchased from Cell Signaling Technology Inc. (Danvers, MA, USA), anti-KDM1/LSD1 (ab129195), anti-CD9 (ab223052), anti-Annexin V/ANXA5 (ab108194), anti-Osterix (ab22552) and anti-CD3

(ab16669) antibodies from Abcam (Cambridge, MA, USA), an anti-ALP antibody (PAB12279) from Abnova (Taipei, Taiwan), an anti-SDF-1 antibody (17402-1-AP) from proteintech (Rosemond, IL, USA), an anti-F4/80 antibody (MCA497R) from Bio-Rad Laboratories (Hercules, CA), and an anti-TRAP antibody (sc-30833) from Santa Cruz Biotechnology (Santa Cruz, CA, USA). RPMI 1640, penicillin–streptomycin, and a protease inhibitor were obtained from FUJIFILM Wako Chemicals. (Osaka, Japan). Collagenase I, TRIzol, and α-MEM were supplied by Thermo Fisher Scientific, Inc. (Waltham, MA, USA). One collagenase I unit liberated 1 μmol of L-leucine equivalents from collagen in 5 h at 37˚C, pH 7.5. Fetal bovine serum (FBS) was obtained from Sigma (St Louis, MO) and was heat inactivated at 56˚C for 30 min.

## Animals

C57BL/6J mice were purchased from CLEA Japan (Tokyo, Japan). Animal experiments were performed in accordance with the guidelines of ARRIVE and the institutional rules for the use and care of laboratory animals at Kindai University. All experimental procedures on animals were approved by the Experimental Animal Welfare Committee of Kindai University (permit number: KAME-2022-073).

## Cell culture

The mouse bone marrow-derived stromal cell line, ST2 (RIKEN, Tsukuba, Japan) was cultured in RPMI 1640 with 10% FBS and 1% penicillin-streptomycin. ST2 cells were differentiated into osteoblastic cells with 200 ng/mL BMP-2 (FUJIFILM Wako Chemicals) for 3 days. Mouse osteoblasts were isolated from the neonatal calvariae of 3- to 5-day-old C57BL/6J male mice as previously described [29]. Briefly, the calvariae of neonatal mice were digested 4 times with 1 mg/mL collagenase I and 0.25% trypsin at 37˚C for 20 min with gentle agitation. Cells from the second, third, and fourth digestions were collected. The resultant cells were cultured in α-MEM supplemented with 10% FBS and 1% penicillin/streptomycin. Isolated osteoblasts exhibited ALP activity and formed mineralized nodules. Osteoblasts at passage 2 were used in further experiments. Cells were cultured at 37˚C in a humidified atmosphere containing 5% $CO_2$.

## Isolation of CREVs

EV-depleted FBS was prepared by ultracentrifugation at 130,000×$g$ at 4˚C for 16 h. CREVs were collected from the mineralized nodule-formed mouse osteoblast culture by the sequential centrifugation method following matrix digestion. A total of $1 \times 10^6$ mouse osteoblasts were seeded on a 10-cm culture plate and cultured in α-MEM with 10% EV-depleted FBS and 1% penicillin/streptomycin at 37˚C for 5 days. Culture medium was changed to α-MEM with 10% EV-depleted FBS, 10 mM β-glycerophosphate, and 1% penicillin/streptomycin, and osteoblasts were cultured for 14 days. After the formation of mineralized nodules had been confirmed, culture medium was removed and osteoblasts were washed with PBS twice. Following the addition of 500 units/mL collagenase I to the cell culture, the CREV-containing matrix was digested at 37˚C for 30 min. Collagenase solution containing osteoblasts and CREVs was centrifuged at 300×$g$ at 4˚C for 5 min to separate osteoblasts from the suspension, centrifuged at 3,000×$g$ at 4˚C for 20 min to remove cell debris, filtered using a Stericup filter unit with a 0.22-μm PVDF filter to purify CREVs, and ultracentrifuged at 130,000×$g$ at 4˚C for 70 min with a Himac CP80NX system (HITACHI, Tokyo, Japan) to isolate CREVs (pellet). The supernatant was removed, and the pellet was washed with PBS once and then ultracentrifuged again. The pellet was resuspended in fresh PBS and stored at −80˚C until used. The total protein level was measured with a BCA Protein Assay Kit (Pierce, Rockford, IL). The size distribution of CREVs was analyzed using a NanoSight LM10V-HS system (Malvern Instruments, Malvern, UK).

## Western blotting

Mouse osteoblasts and CREVs were lysed using RIPA buffer with protease inhibitors, and were centrifuged at 12,000×$g$ at 4°C for 10 min. Supernatants were collected as protein samples. After protein levels were quantified using a BCA assay reagent, aliquots with an equal amount of protein (5 μg) were separated on a 4–20% Mini-PROTEAN Tris-Glycine eXtended Precast Protein Gel (Bio-Rad Laboratories) at 150 V for 60 min, and then transferred onto PVDF membranes at 45 V for 90 min. Membranes were blocked with 3% skim milk in Tris-buffered saline/0.05% Tween 20 (TBS-T) at room temperature for 60 min and washed with TBS-T. Membranes were incubated with primary antibodies against KDM1/LSD1 (1:5000), β-actin (1:1000), CD9 (1:1000), and Annexin A5 (1:10000) at 4°C overnight and then washed with TBS-T. Membranes were incubated with a secondary antibody (anti-rabbit IgG-HRP antibody, 1:10000) at room temperature for 60 min and then washed with TBS-T. Immune complexes on membranes were visualized using ECL Western blot analysis Detection Reagent (GE Healthcare, Tokyo, Japan) and analyzed with Amersham Imager 600 (GE Healthcare).

## ALP activity

Collagenase-treated mouse osteoblasts and CREVs were lysed with RIPA buffer after the CREV isolation process. Lysed cells and CREVs were centrifuged at 12,000×$g$ at 4°C for 10 min. Supernatants were collected as protein samples. Mouse osteoblasts were cultured with or without 4 μg/mL CREVs for 72 h, cells in 24-well plates were rinsed three times with PBS, and 200 μL of distilled water was then added to each well. Lysed cell homogenates were centrifuged at 12,000×$g$ at 4°C for 10 min. Supernatants were collected as protein samples. After protein levels were quantified using BCA assay reagent, protein samples were diluted to 10 or 1 μg protein/mL and analyzed with LabAssay ALP (FUJIFILM Wako Chemicals) according to the manufacturer's protocol.

## Preparation of cationized gelatin hydrogel sheets

Cationized gelatins were synthesized by converting the carboxyl groups of gelatin with the amino groups, previously described [30]. Briefly, 7.8 g ethylenediamine (EDA) was added into 250 ml of gelatin solution (25 mg/ml) in 0.1 M phosphate-buffered solution (PB, pH = 5.0) at 40°C. The pH of the solution was adjusted to 5.0 by adding 11 M HCl aqueous solution. 5.35 g ethylenedichloride (EDC) was added into the solution and PB was added into the solution to give the final volume of 500 ml. After stirring at 40°C for 18 h, the gelatin solution was dialyzed against double-distilled water (DDW) for 3 days at room temperature. The dialyzed solution was freeze-dried to obtain cationized gelatins. To determine the percentage of amino groups introduced into gelatin, the conventional 2,4,6-trinitrobenzene sulfonic acid method was performed [31]. The percentage was 19.9 mole% per the carboxyl groups of gelatins. 1.2 ml of cationized gelatin aqueous solution (5.0 mg/ml) was poured into polytetrafluoroethylene mold (43 mm × 43 mm) and freeze-dried to prepare the cationized gelatin hydrogel sheets. The hydrogel sheets were stabilized by dehydrothermal crosslinking at 160°C for 72 h with dry oven (AURORA DN-305, Sato Vacuum Inc., Japan). Gelatin hydrogel sheets were punched out to circular disc with the diameter of 1.5 mm for the transplantation. 1.5 μL of CREVs suspension in PBS (6.67 μg/μL) was soaked into disc-shaped gelatin hydrogel.

## Bone defect model

Twelve-week-old female C57BL/6 mice were divided into two groups: vehicle (n = 8) and CREVs (n = 8). A femoral bone defect was induced in mice using a previously described

method with some modifications [29]. Briefly, an incision of 5 mm in length was made in the anterior skin of the central femur of the right leg under 2% isoflurane anesthesia. The muscle was split to expose the surface of the femoral bone and a hole with a diameter of 0.8 mm was created in the femur using a drill. The hole was irrigated with saline to prevent thermal necrosis. We applied a cationized gelatin hydrogel as a sustained release carrier of CREVs at the damaged site of the femur [32]. Regarding the local administration of CREVs, single disc-shaped gelatin hydrogel sheets with a diameter of 1.5 mm were loaded with CREVs (10 µg) or PBS and placed in the hole of the femoral defect. Incised skin was then sutured in a sterile manner, and anesthesia was discontinued. New bone started to form, and ALP-positive cells were observed at the damaged area 7 days after the femoral bone defect, as previously reported [33]. Therefore, we selected the time point of 7 days after the bone defect to assess bone repair in the present study.

## Quantitative computed tomography (qCT) analysis

Mice were anesthetized with 2% isoflurane, and femurs were scanned using Cosmo Scan GX II (Rigaku Corporation, Tokyo, Japan). The parameters used for CT scans were as follows: tube voltage of 90 kV, tube current of 88 µA, and isotropic voxel size of 25 µm. The area of bone damage in each femur was quantified with an image-processing program (ImageJ, http://rsbweb.nih.gov/ij/download.html). The ratio of the bone volume to tissue volume (BV/TV) within the bone defect region was calculated to evaluate new bone formation in the bone defect with an image-processing program using Analyze 14.0 (AnalyzeDirect, Inc., KS, USA)

## Histological analysis

Mice were anesthetized with 2% isoflurane 7 days after the femoral bone defect. We collected femurs after fixation with 4% paraformaldehyde perfusion under anesthesia with 2% isoflurane. The femur was fixed with 4% paraformaldehyde for 24 h, demineralized in a 22.5% formic acid and 340 mM sodium citrate solution for 24 h, and then embedded in paraffin. Four-micrometer-thick sections were obtained. Histological analyses were performed as previously described [29]. Deparaffined sections of the femoral bone defect were stained with hematoxylin/eosin, Alcian blue, or Toluidine blue. To calculate osteoblastic or osteoclastic cells in the bone defect area, deparaffined sections of the femoral bone defect were incubated with the anti-ALP antibody at a dilution of 1:300, the anti-Osterix antibody at a dilution of 1:500, the anti-F4/80 antibody at a dilution of 1:1000, the anti-CD3 antibody at a dilution of 1:100, the anti-PDGFRα antibody at a dilution of 1:500, the anti-SDF-1 antibody at a dilution of 1:100, or the anti-TRAP antibody at a dilution of 1:100, respectively, followed by an incubation with an appropriate horseradish peroxidase-conjugated secondary antibody. Positive signals were visualized using the tyramide signal amplification system (PerkinElmer, Waltham, MS, USA), and sections were counterstained with 4′,6-diamidino-2-phenylindole (DAPI). Sections were observed under a fluorescence microscope (BZ-710, Keyence, Osaka, Japan). The metachromatic areas of Toluidine blue-stained sections, the Alcian blue-stained area, and the numbers of ALP-positive cells and TRAP-positive cells in the bone defect area were measured using ImageJ in a blinded manner.

## qRT-PCR

Total RNA was prepared from cells cultured with 4 µg/mL CREVs using TRIzol reagent according to the manufacturer's protocol. Reverse transcription was performed using a Prime-Script RT reagent Kit with a gDNA eraser (Takara Bio, Kyoto, Japan), and real-time PCR was performed using the SYBR Premix Ex Taq™ II kit (Takara Bio) for 40 cycles of a two-step PCR

**Table 1. Sequences of primers for qRT-PCR.**

| ALP | Forward | 5′-CGGATAACGAGATGCCACCA-3′ |
|---|---|---|
| | Reverse | 5′-GCCATCTAGCCTTGTACCCC-3′ |
| Col1a1 | Forward | 5′-AACCCTGCCCGCACATG-3′ |
| | Reverse | 5′-CAGACGGCTGAGTAGGGAACA-3′ |
| Gapdh | Forward | 5′-AGGTCGGTGTGAACGGATTTG-3′ |
| | Reverse | 5′-GGGGTCGTTGATGGCAACA-3′ |
| Osteocalcin | Forward | 5′-CCTGAGTCTGACAAAGCCTTCA-3′ |
| | Reverse | 5′-GCCGGAGTCTGTTCACTACCTT-3′ |
| Osterix | Forward | 5′-AGCGACCACTTGAGCAAACAT-3′ |
| | Reverse | 5′-GCGGCTGATTGGCTTCTTCT-3′ |

*ALP*: Alkaline phosphatase, *Col1a1*: Type 1 Collagen, *Gapdh*: Glyceraldehyde-3-phosphate dehydrogenase

amplification (95˚C for 3 s and 60˚C for 30 s) on an Applied Biosystems Step One Plus™ Real-Time PCR System (Thermo Fisher Scientific). The mRNA levels of target genes were normalized against glyceraldehyde-3-phosphate dehydrogenase (Gapdh) as an internal standard. Primer sequences (forward and reverse) are listed in Table 1.

## Mineralization assay

Mouse osteoblasts were cultured in α-MEM supplemented with 10% FBS and 1% penicillin–streptomycin. After reaching confluency, cells were cultured with 10 mM β-glycerophosphate and 50 μg/mL ascorbic acid with or without 4 μg/mL osteoblast-derived CREVs for 3 weeks. Cells fixed with ice-cold 70% ethanol and stained with Alizarin red to detect mineralization. Regarding quantification, cells stained with Alizarin red were treated with cetylpyridinium chloride (FUJIFILM Wako Chemical), the extracted stain was transferred to a 96-well plate, and absorbance at 570 nm was measured.

## Statistical analysis

The sample size for each experiment was selected using G*Power 3.1 software (https://gpower.software.informer.com/3.1/). We used parameters based on our previous studies for a power analysis. All data were expressed as the mean ± the standard error of the mean (SEM). The significance of differences was evaluated using a one-way ANOVA followed by the Tukey–Kramer post hoc test for multiple comparisons. The significance of differences was evaluated using the Mann–Whitney $U$ test and unpaired $t$-test for comparisons of two groups *in vivo* and *in vitro*, respectively. The significance level was set at $p < 0.05$. All statistical analyses were performed using GraphPad PRISM 9 software (La Jolla, CA).

## Results

### Collection of matrix vesicles secreted from osteoblasts

EVs released from the mineralized nodule-formed osteoblast culture with the collagenase treatment (CREVs) were used as putative MtVs in the present study according to a previously reported method [24, 25]. CREVs were collected from mineralized nodule-formed mouse osteoblast cultures by the sequential centrifugation method following matrix digestion (Fig 1A). The particle sizes of most CREVs were <200 nm regardless of the 0.22-μm filtering process (Fig 1B). Therefore, we used 0.22-μm filtered CREVs in experiments. CREV samples were positive for the exosome marker protein CD9, while the nuclear marker protein KDM1/LSD1 and

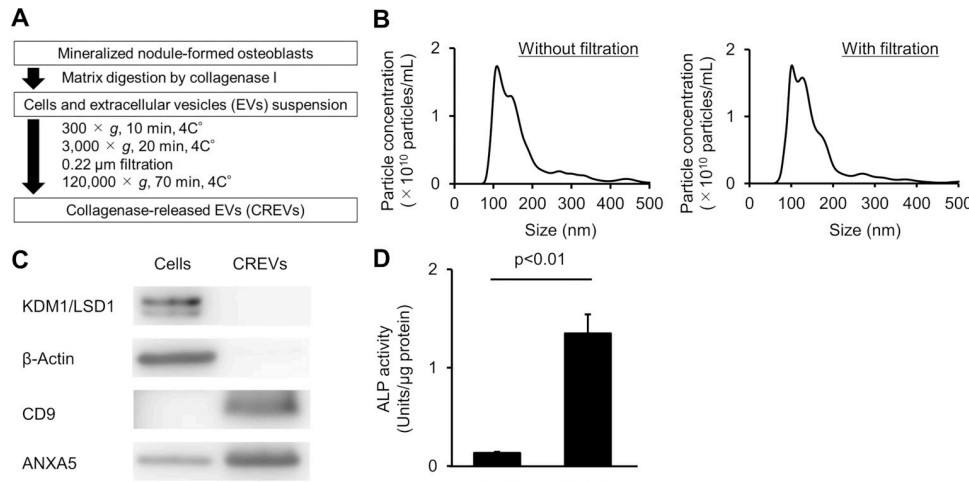

**Fig 1. Analyses of collagenase-released extracellular vesicles isolated from mouse osteoblasts (CREVs).** (A) A schema of a preparation of CREVs. (B) The particle size and concentration of CREVs in NanoSight meter analyses. Results are expressed as the mean of 5 measurements within one experiment in each group. (C) Western blotting analysis of KDM1/LSD1, β-actin, CD9, and Annexin A5 (ANXA5) in mouse osteoblast cell lysates and CREVs. (D) ALP activities of mouse osteoblast cell lysates and CREVs. Results are expressed as the mean ± SEM of 4 independent experiments. Statistical analyses were performed using the unpaired *t*-test.

the cytoplasm marker β-actin were not detected (Fig 1C), suggesting that CREVs included exosome-like particles with negligible contamination by cell debris. MtVs expressed Annexin A5, a calcium transporter, and exhibited high ALP activity (Fig 1C and 1D), which are crucial for the formation of hydroxyapatite [34]. Moreover, CREVs showed higher Annexin A5 expression and ALP activity than mouse osteoblast cell lysates. These results suggest that the CREVs collected possessed MtV-specific characteristics.

## Effects of the local administration of CREVs on bone repair

We examined the effects of the local administration of CREVs on bone repair after a femoral bone defect in mice. CREVs were administrated locally in a gelatin hydrogel as a sustained release carrier to mice at the damaged site after a femoral bone defect. Bone defect areas were significantly smaller than those in vehicle-administered mice 7 days after the femoral bone defect (Fig 2A and 2B). The BV/TV of new bone at bone defect sites was significantly increased by the local administration of CREVs 7 days after the femoral bone defect (Fig 2C). HE sections of bone defect sites showed that new bone tissue was increased by the local administration of CREVs (Fig 2D).

## Analysis of osteogenesis and chondrogenesis at damaged sites

Histological analyses of damaged sites after the femoral bone defect were performed to examine the effects of the local administration of CREVs on bone formation. The numbers of ALP-positive cells, which are putative osteoblasts, per bone defect area, the bone surface, and the number of DAPI-positive nucleus were significantly increased at the damaged site by the local administration of CREVs 7 days after the femoral bone defect (Fig 3A–3C). The local administration of CREVs did not affect the number of TRAP-positive multinucleated cells (MNCs) 7 days after the femoral bone defect (Fig 3D). In sections stained with Toluidine blue and Alcian blue, metachromatic and stained areas, indicating the cartilage matrix, were significantly

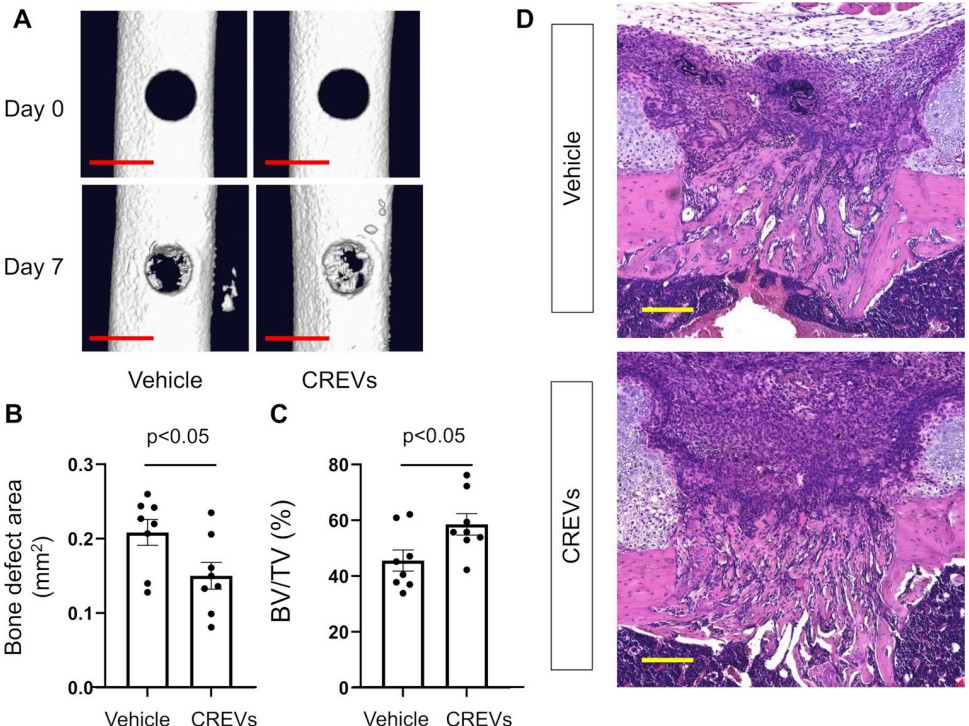

**Fig 2. Effects of the local administration of CREVs on bone repair after a femoral bone defect.** (A) Three-dimensional qCT images of damaged sites 7 days after a femoral bone defect. The scale bar indicates 1 mm. (B) The bone defect area at the damaged site 7 days after the femoral bone defect. (C) The bone volume/tissue volume (BV/TV, %) of mineralized bone that formed in the hole region 7 days after the femoral bone defect. Results are expressed as the means ± SEM of 8 mice per group. Statistical analyses were performed using the Mann–Whitney *U* test (B, C). (D) Images of HE-stained sections of damaged sites 7 days after the femoral bone defect. Scale bars indicate 200 μm.

increased by the local administration of CREVs 7 days after the femoral bone defect (Fig 4A and 4B).

## Analysis of periosteum site

Area of larger number of cell accumulation was observed at periosteum site 7 days after the femoral bone defect with CREVs administration, compared to vehicle group (Fig 2, S1 Fig). Numerous PDGFRα-positive cells (skeletal stem/progenitor cells (SSPCs)) and Osterix-positive cells (preosteoblastic cells) were observed at periosteum area 7 days after the femoral bone defect with CREVs transplantation (S2 and S3 Figs), although few ALP-positive cells (osteoblastic cells), F4/80-positive cells (macrophages), TRAP-positive cells (osteoclasts) and CD3-positive cells (T-lymphocytes) were detected (S2 Fig). In addition, the number of SDF-1-positive cells seemed to be increased at the periosteum area by the local administration of CREVs 7 days after the femoral bone defect (S4 Fig).

## Effects of CREVs on osteoblast differentiation *in vitro*

Since MtVs enhanced bone repair in mice, we speculated that they may affect osteoblastic differentiation by directly acting on osteoblastic cells. Therefore, we examined the effects of CREVs on the osteoblastic differentiation of mouse mesenchymal ST2 cells and the phenotypes of mouse osteoblasts *in vitro*. The CREV treatment significantly reduced BMP-2-induced increases in the mRNA levels of osterix and osteocalcin, but not those of ALP and type I

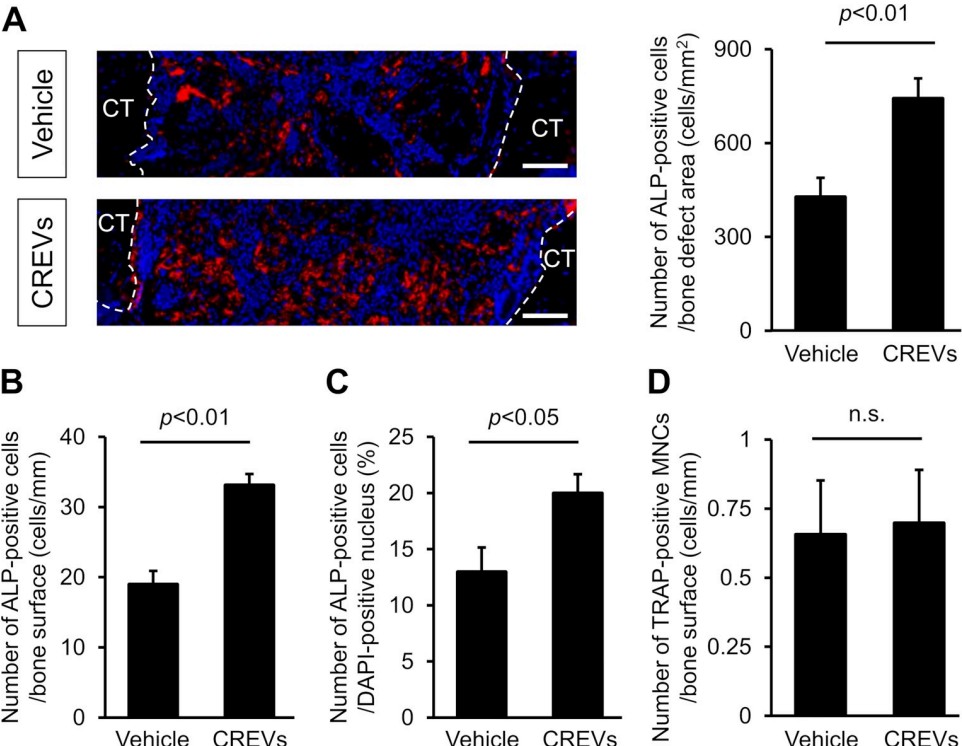

**Fig 3. Effects of the local administration of CREVs on the osteogenic marker at damaged sites.** Images of ALP-positive cells and the number of ALP-positive cells per area ($mm^2$) (A), bone surface (mm) (B), and the number of DAPI-positive nucleus (C) at the damaged site 7 days after the femoral bone defect. Results are expressed as the means ± SEM of 8 mice per group. Scale bars indicate 100 μm. The dotted line indicates the boundary with cortical bone. (D) The number of TRAP-positive multinucleated cells (MNCs) per bone surface (mm) at the damaged site 7 days after the femoral bone defect. Results are expressed as the means ± SEM of 6 (Vehicle) or 8 (CREVs) mice per group, respectively. Statistical analyses were performed using the Mann–Whitney *U* test.

collagen in ST2 cells (Fig 5). The CREV treatment significantly decreased the mRNA levels of osterix, osteocalcin, and type 1 collagen, but not ALP, in mouse osteoblasts (Fig 6A). On the other hand, the CREV treatment did not affect ALP activity or mineralization with the Alizarin red stain in mouse osteoblasts (Fig 6B and 6C).

## Discussion

In the present study, CREVs were positive for the MtV marker Annexin 5A and exhibited higher ALP activity than osteoblast cell lysates, indicating that the CREVs examined were putative MtVs. The majority of EVs are classified into three main subtypes of vesicles based on their vesicle formation mechanism: exosomes, microvesicles, and apoptotic bodies [7, 8]. Exosomes with a diameter of 50–150 nm are formed by the endosomal route [35]. Microvesicles with a diameter of 50–1000 nm are formed by budding from the plasma membrane [36]. Apoptotic bodies with a diameter of 500–2000 nm are formed by plasma membrane blebbing [37]. An electron micrograph study of bone tissue showed that MtVs are EVs with a diameter of 100–300 nm, and their secretion is considered to involve a microvesicle-like mechanism, budding from the plasma membrane [17]. In the present study, analyses using nanoparticle tracking revealed that the maximum peak of the diameter population was 100–150 nm, with few EVs having a diameter >200 nm. Moreover, CREVs were positive for the exosome marker CD9. These results suggest that CREVs contain exosome-like EVs as well as a few

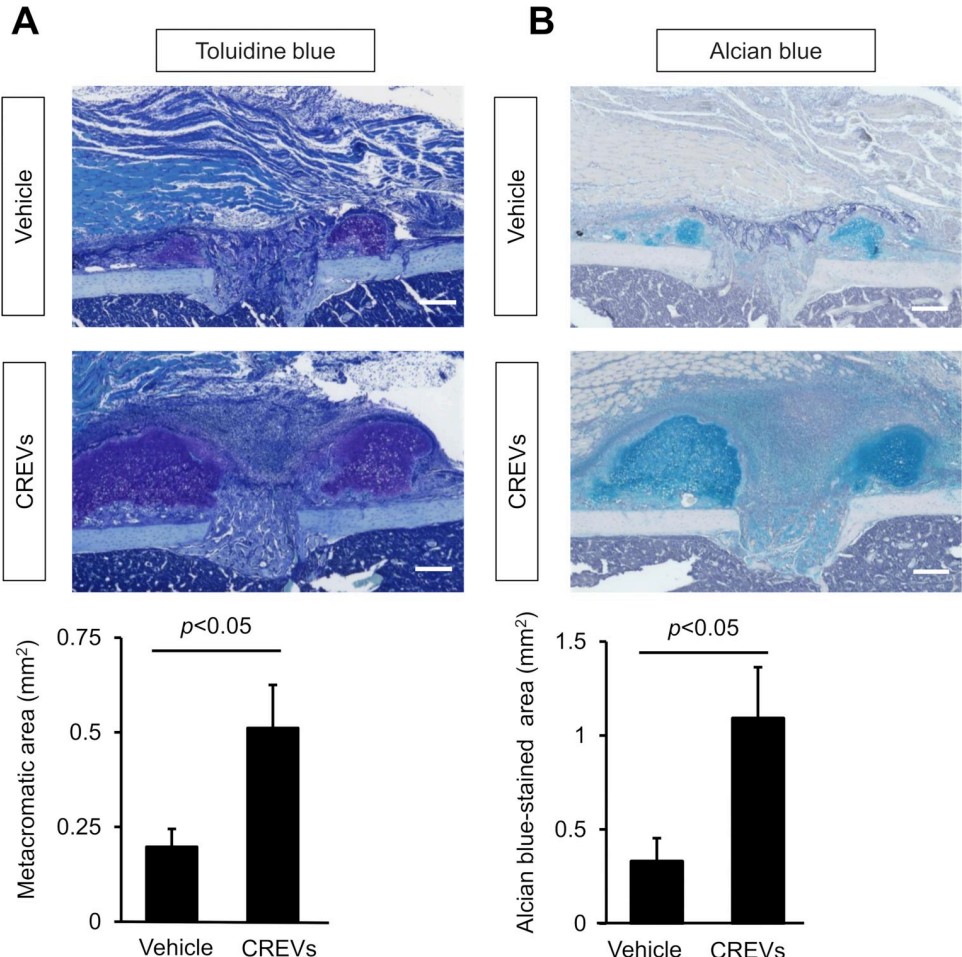

**Fig 4. Effects of the local administration of CREVs on the chondrogenic marker at damaged sites.** (A) Images of Toluidine blue-stained sections and the area of the metachromatic-stained region at the damaged site 7 days after the femoral bone defect. Results are expressed as the means ± SEM of 8 mice per group. (B) Images of Alcian blue-stained sections and the area of the Alcian blue-stained region at the damaged site 7 days after the femoral bone defect. Results are expressed as the means ± SEM of 6 (Vehicle) or 8 (CREVs) mice per group, respectively. Scale bars indicate 200 μm. Statistical analyses were performed using the Mann–Whitney $U$ test.

microvesicle-like EVs. Recent findings indicated the intracellular formation of MtVs via an exosome-like mechanism [38–40].

Cationized gelatin hydrogels are negatively charged molecules due to degradation, and EVs possess a negative charge on their surface; therefore, cationized gelatin hydrogels are used as their carrier for local administration [33, 41]. Transplanted gelatin hydrogels were observed at the damaged site 7 days after the femoral bone defect in the present study. Moreover, we previously reported that myoblast-derived EV-containing gelatin hydrogels were degraded 9 days after transplantation at the site of the femoral bone defect [33]. These findings suggest that gelatin hydrogels are retained at damaged sites in femoral bone, and osteoblast-derived CREVs are released for 7 days.

In the present study, we showed that the local administration of CREVs in a gelatin hydrogel enhanced bone repair after a femoral bone defect in mice, indicating that MtVs possess the ability to induce bone repair after bone injury in mice. However, Uenaka *et al.* reported that the local administration of mouse osteoblast-derived EVs collected from a mixture of culture

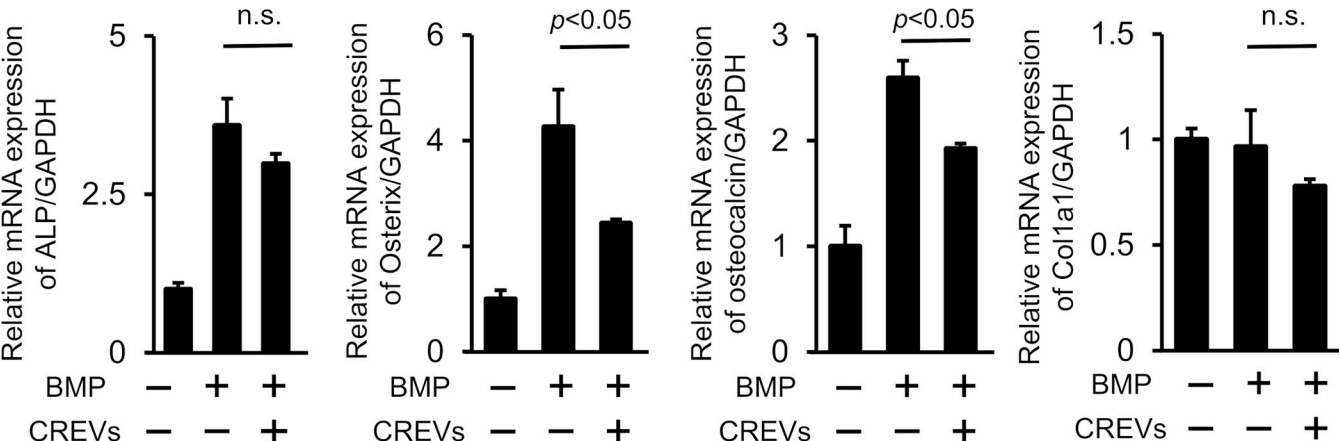

**Fig 5. Effects of CREVs on the osteoblastic differentiation of mouse mesenchymal ST2 cells.** The mRNA expression of osteogenic markers in ST2 cells cultured with or without 200 ng/mL BMP-2 and/or 4 µg/mL CREVs for 72 h. Results are expressed relative to Gapdh mRNA levels and are shown as the means ± SEM of 4 independent experiments. Statistical analyses were performed using a one-way ANOVA followed by the Tukey–Kramer post hoc test.

medium and enzymatic-digested cells inhibited calvaria bone repair in mice [13]. Although the mechanisms responsible for the different effects of osteoblast-derived EVs and MtVs on bone repair remain unclear, they may involve differences in EV characteristics, the conditions of osteoblasts, or EV-transplanted sites. Uenaka et al. suggested that bone repair by osteoblast-

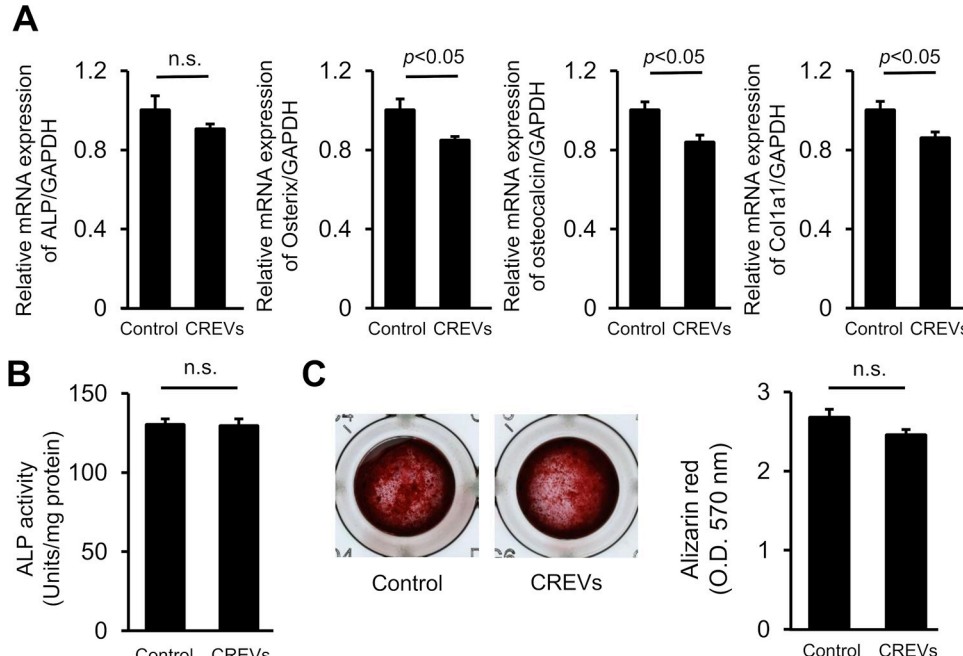

**Fig 6. Effects of CREVs on the phenotypes of mouse osteoblasts.** (A) The mRNA expression of osteogenic markers in mouse osteoblasts cultured with or without 200 ng/mL BMP-2 and/or 4 µg/mL CREVs for 72 h. Results are expressed relative to Gapdh mRNA levels and are shown as the means ± SEM of 4 independent experiments. (B) The ALP activity of mouse osteoblasts cultured with or without 4 µg/mL CREVs for 72 h. Results are expressed as the mean ± SEM of 4 independent experiments. (C) Images of Alizarin red-stained mineralized mouse osteoblasts cultured with 50 µg/mL ascorbic acid and 10 mM β-glycerophosphate in the presence or absence of 4 µg/mL CREVs. The absorbance of Alizarin red extracted with cetylpyridinium chloride solution (570 nm) is expressed as the mean ± SEM of 8 independent experiments. Statistical analyses were performed using the unpaired t-test.

derived EVs with a diameter >200 nm was delayed in mice [13]. The present results indicated that CREVs with a diameter <200 nm promoted bone repair after a femoral bone defect in mice. Therefore, we speculate that EVs with a diameter <200 nm and high affinity for the bone matrix may be crucial for bone repair and regeneration activity. Furthermore, the characteristics of EVs may be affected by a collagenase treatment during collection. EVs collected from the leg bones of 17-day-old chicken embryos without a collagenase treatment exhibited weaker ALP activity and poorer mineralizing properties than those with the collagenase treatment [42]. Therefore, EVs collected from culture medium appeared to be a different subtype of the EV population to CREVs. In addition, Uenaka *et al.* collected EVs from short-term mineralized osteoblasts (2 days) and CREVs from long-term mineralized osteoblasts (14 days). The estimation of well-mineralized osteoblast-derived EVs may be important for clarifying osteoblast-derived EV effects on bone regeneration. Alternatively, Uenaka *et al.* used a calvaria bone defect model to estimate osteoblast-derived EV effects on bone regeneration; however, CREVs were administered to the femoral bone defect in the present study. Since the regeneration of calvaria bone and femoral bone predominantly proceeds with membranous ossification and endochondral and membranous ossification processes, respectively, osteoblast-derived EV effects on bone repair may differ depending on the damaged site.

The present study revealed that the local administration of CREVs significantly increased the number of ALP-positive cells at the damaged site after the femoral bone defect in mice, indicating that MtVs enhance bone repair by increasing osteoblastic bone formation. Moreover, we revealed that the local administration of CREVs significantly enhanced chondrogenesis at the damaged site after the femoral bone defect in mice. These results suggest that osteoblastic differentiation and chondrogenesis are both involved in enhancing the effects of MtVs on bone repair after a femoral bone defect in mice. Although the mechanisms by which MtVs enhance osteogenesis and chondrogenesis have not yet been elucidated in detail, CREVs did not affect the osteogenic differentiation of ST2 cells or mouse osteoblasts when they were directly added to the cell culture medium *in vitro*. Moreover, the addition of CREVs to the cultured medium did not affect ALP activity or mineralization in mouse osteoblasts. These results suggest that MtVs did not affect osteogenic differentiation through direct effects on osteoblasts or mesenchymal cells. Bone repair proceeds in three phases: inflammation, renewal, and remodeling. The cells recruited to the damaged site in bone in the inflammation phase induce the migration of osteogenic cells. For example, macrophages are recruited and induce the differentiation of mesenchymal stem cells into osteoblasts by secreting transforming growth factor-β, insulin-like growth factor-1, and fibroblast growth factor 2 [43]. Therefore, MtVs may promote bone repair through cells other than osteoblasts or mesenchymal cells.

SSPCs are cells with the characteristics of mesenchymal stem cells as well as mobilized from bone marrow, periosteum and skeletal muscles during bone regeneration [44]. SSPCs possess different regenerative potential according to their origin. Namely, SSPCs from bone marrow, periosteum and muscles contribute to osteoblastic bone formation, osteochondral bone formation and cartilage formation, respectively [45–47]. Since the local administration of CREVs seemed to increase numbers of PDGFRα- and Osterix-positive cells as well as cartilage/bone formation at the bone damaged sites in our study, CREVs might recruit SSPCs from bone marrow, periosteum and skeletal muscles, then improving bone regeneration *in vivo*. Our data showed that numerous SDF-1-positive cells were observed at the periosteal area of the damaged site with the local administration of CREVs. SDF-1 is a chemokine involved in the recruitment of mesenchymal stem cells at the fracture sites [48–51]. Taken together, the local administration of CREVs might recruit PDGFα-positive SSPCs at the bone damaged sites partly through SDF-1, leading to an enhancement of chondrogenic/osteoblastic bone formation and subsequent bone repair in mice. Alternatively, MtVs may enhance bone repair

through the supply of hydroxyapatite for mineralization. Further studies are needed to clarify this issue.

There are some limitations that need to be addressed. We cannot completely rule out the possibility that osteoblast-derived CREVs exert different effects on the bone repair process at different time points after bone injury because μ-CT and histological analyses were performed 7 days after the femoral bone defect. Furthermore, μ-CT scanning with a voxel size of 25 μm may be low to analyze microarchitectural data. μ-CT scanning with a higher resolution will provide insights into the mechanisms by which osteoblast-derived CREVs enhance bone repair. Moreover, in the *in vitro* study, the effects of osteoblast-derived CREVs on osteoblastic differentiation were estimated using ST2 cells. Experiments using primary mesenchymal stem cells may be necessary to clarify the effects of osteoblast-derived CREVs on the differentiation of mesenchymal stem cells into osteoblasts *in vitro* because ST2 cells may not represent mesenchymal stem cells. A previous study reported that osteoblast-derived EVs exhibited biological activity at concentrations of 1–2.5 μg/mL [16, 24]. Although osteoblast-derived CREVs were used at a concentration of 4 μg/mL in the *in vitro* study without significant osteogenic effects, the concentration of osteoblast-derived CREVs may have been too low to exert significant osteogenic effects. Another limitation is that in the *in vivo* study, we used a gelatin hydrogel as the osteoblast-derived CREV carrier and cellular scaffold. Therefore, a 2-dimensional culture system may be insufficient to estimate the response of osteogenic cells to osteoblast-derived CREVs. In addition, most of our data were observational and lacking in mechanical insights. Therefore, further studies are necessary to clarify the mechanisms by which osteoblast-derived CREVs enhance bone repair.

The administration of osteoblast-derived EVs is expected to be applied to bone regeneration therapy instead of cell transplantation [52, 53]. In contrast to cell transplantation, EVs do not continue to be sustained and change their characteristics in the body after their administration. In conclusion, we herein demonstrated that the local administration of MtVs with a gelatin hydrogel enhanced bone repair partly through osteogenesis and chondrogenesis at the damaged site after a femoral bone defect in mice. The present results suggest the potential of MtVs for bone regeneration therapy.

## Supporting information

**S1 Fig. Images of HE-stained sections at periosteal site 7 days after the femoral bone defect.** An expanded image of red square region in the center image was presented at right panel. Scale bars in left, center, and right images indicate 500, 500, and 50 μm, respectively. (TIF)

**S2 Fig. Images of ALP-, Osterix-, F4/80-, TRAP-, and CD3-positive cells or HE-stained sections at periosteal site 7 days after the femoral bone defect with CREVs transplantation.** Scale bars indicate 200 μm. The white and yellow line indicates the boundary with cortical bone (CT) and cartilage (CL), respectively. (TIF)

**S3 Fig. Images of PDGFRα-positive cells at the periosteal site 7 days after the femoral bone defect.** Scale bars indicate 200 μm. The white and yellow line indicates the boundary with cortical bone (CT) and cartilage (CL), respectively. (TIF)

**S4 Fig. Images of SDF-1-positive cells at the periosteal site 7 days after the femoral bone defect. Scale bars indicate 200 μm.** The white and yellow line indicates the boundary with

cortical bone (CT) and cartilage (CL), respectively.
(TIF)

**S1 Raw images.**
(TIF)

## Author Contributions

**Data curation:** Yuya Mizukami, Naoyuki Kawao, Yoshimasa Takafuji, Takashi Ohira, Kiyotaka Okada.

**Formal analysis:** Yuya Mizukami.

**Funding acquisition:** Yuya Mizukami, Yoshimasa Takafuji, Hiroshi Kaji.

**Investigation:** Yuya Mizukami, Naoyuki Kawao, Yoshimasa Takafuji, Takashi Ohira, Kiyotaka Okada.

**Methodology:** Yuya Mizukami.

**Project administration:** Yuya Mizukami.

**Resources:** Jun-Ichiro Jo, Yasuhiko Tabata.

**Supervision:** Hiroshi Kaji.

**Visualization:** Yuya Mizukami.

**Writing – original draft:** Yuya Mizukami.

**Writing – review & editing:** Naoyuki Kawao, Yoshimasa Takafuji, Takashi Ohira, Kiyotaka Okada, Jun-Ichiro Jo, Yasuhiko Tabata, Hiroshi Kaji.

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
