## [Decision Letter · Decision Letter 0]

6 Dec 2022

PONE-D-22-31340Matrix vesicles promote bone repair after a femoral bone defect in mice.PLOS ONE

Dear Dr. Kaji,

Thank you for submitting your manuscript to PLOS ONE. After careful consideration, we feel that it has merit but does not fully meet PLOS ONE’s publication criteria as it currently stands. Therefore, we invite you to submit a revised version of the manuscript that addresses the points raised during the review process.

We look forward to receiving your revised manuscript.

Kind regards,

Isha Mutreja

Academic Editor

PLOS ONE

Journal Requirements:

Reviewers' comments:

Reviewer's Responses to Questions

**Comments to the Author**

1. Is the manuscript technically sound, and do the data support the conclusions?

Reviewer #1: No

Reviewer #2: Partly

Reviewer #3: No

2. Has the statistical analysis been performed appropriately and rigorously? 

Reviewer #1: Yes

Reviewer #2: Yes

Reviewer #3: I Don't Know

3. Have the authors made all data underlying the findings in their manuscript fully available?

Reviewer #1: Yes

Reviewer #2: Yes

Reviewer #3: Yes

4. Is the manuscript presented in an intelligible fashion and written in standard English?

Reviewer #1: No

Reviewer #2: No

Reviewer #3: Yes

5. Review Comments to the Author

Reviewer #1: Overall Consideration:

In the manuscript, “Matrix vesicles promote bone repair after a femoral bone defect in mice”, the authors utilize evaluate the effects of osteoblast-derived collagenase-released extracellular vesicles on cortical bone regeneration. The author utilized microCT and histology to assess bone healing within this model. This in vivo experiment in couple with in vitro analyses of osteoblast differentiation and mineralization to define mechanism. The authors argue that CREVs enhance healing within this model by increasing chondrogenesis that facilitates bone regeneration (e.g., endochondral ossification). This is an important topic of study, but there are some significant limitations. Overall, the depth and rigor of the study represent the largest weaknesses. A point-by-point review is listed below:

Major comments:

1. Most of the study is observational, and the bit of mechanistic data presented by the authors are negative data.

2. The voxel size (25 microns) utilized in the microCT scanning is large and may be adding artifact to the authors' analyses. Moreover, this limits the ability of the authors to collect microarchitectural data that may point to mechanism(s).

3. Did the authors euthanize mice prior to collection of the femora for analyses?

4. Can the authors demonstrate retainment of the applied scaffold applied within the defect following surgery?

5. Bone healing is a dynamic process. In the model, the authors are using it takes upwards of month for the defect to heal fully, so the day 7-time point the authors chose may not reflect the entire bone healing process. The authors need to discuss this and other limitations of the study.

6. The cortical bone defect model is supposed to be a model of intramembranous bone regeneration and we would not expect cartilage to form within this model. The authors observe some areas of metachromatic toluidine blue staining, and conclude that cartilage is forming within the defect. Toluidine blue can also stain osteoid in this manner. The authors should perform safranin O / fast green staining or some other more specific method to detect cartilage formation within the defect to substantiate further this claim.

7. Standardized bone histomorphometry (e.g., osteoblasts per bone surface, osteoclasts per bone surface) would also help the authors assess the increased healing observed within their model.

8. The periosteal reaction looks greater in the CREV treated group given the images provided. Can the authors quantify this? This may offer a clue to additional mechanism(s) for the authors.

9. The in vitro data supplied show that CREVs do not enhance osteoblast differentiation in vitro, so the authors conclude that enhanced chondrogenesis must be occurring. There are other explanations for their observations (e.g., altered osteoclastogenesis, increased progenitor recruitment from the periosteum, etc.). Direct data showing that CREVs enhance chondrogenesis and/or chondrocyte maturation (or some other mechanism) are needed to support claims made by the authors.

Minor Comments:

1. The grammar issues within the manuscript need to be addressed.

Reviewer #2: The authors present an interesting article characterizing collagenase-digested EVs and the use of these in a femoral defect model and the impact of these on osteogenesis in vitro. Some claims of the EV identification are debatable, some methodology is missing/unclear, and in vitro assessments should more closely match in vivo application of CREVs in a hydrogel. The following major revisions are requested:

Introduction:

1. The introduction is lacking detail about the specific bone defect the authors are addressing. The authors claim that a bone defect is generally repaired without clinical intervention and later mention that clinical intervention is needed in some instances of critical-sized defect repair. Authors should re-structure their introduction to bone repair with the focus of femoral defect repair, such as current methods and needs for new approaches, and limitations of current approaches for this specific repair problem.

Methods:

2. Collagenase usage is listed as Units/mL. Define these arbitrary ‘units’.

3. ALP activity is missing methods. Based on figures, it seems this assay was performed on the CREV. Add details explaining what samples were used and how much of these samples were used in this assay.

4. Authors utilize a gelatin hydrogel to deliver CREV to the bone defect site; however, there is no methodology on these hydrogels. Authors need to include information on methods of gelatin hydrogel production, characteristics of hydrogels (porosity, mechanics, etc.), and release characteristics of encapsulated EVs. If this is already published work, then this should be added to the introduction explaining these findings.

5. What was the dosing of CREV used in the mineralization assay? Add this to the methods section.

6. Cell isolation- how were osteoblast cells verified for correct cell isolation? Information on cell line manufacturer for ST2 cells is also needed.

Discussion

7. Authors seem to be claiming that CREVs may be secreted in an exosome-like fashion, but in order to claim this, authors must provide more supporting information than CD9+ and similar size range. This sentence could be removed instead.

8. Authors use CREVs and MtEVs interchangeably but note that CREVs may contain exosome-like EVs and MtEVs, but later claim MtEVs induce bone repair based on CREV results. Authors should clarify the identification of their EVs (likely MtEVs) and use this consistently within the manuscript. For clarity, authors should consider only using MtEVs as an acronym for their EV used in this study.

9. Authors assess osteogenesis in a 2D model in vitro but to more accurately represent in vivo work, authors should examine the response of osteogenic cells seeded on hydrogels containing CREVs.

10. There was a lack of mineralization differences in vitro and authors suggest CREVs could work through a different mechanism. Authors should consider examining cell activity in response to CREVs and protein release of those related to osteoclastogenesis and inflammation (OPG, IL6, etc.), especially from mesenchymal stem cells, as this may prove interesting mechanistic results of CREVs.

Reviewer #3: This study by Mizakumi et al. isolates vesicles from mouse osteoblast cultures. Vesicles were used to treat a femoral defect and osteoblast cultures. There are several major concerns with experimental design and methods of the study that make it difficult to interpret the results.

1. There is no indication of the sex or age of the mice used in the femoral defect model experiment. No power analysis is included to indicate how number of animals used was determined.

2. ALP cells in Fig 3 should be quantitated per number of DAPI number of cells.

3. Sections stained with toluidine blue to indicate areas of cartilage should instead be stained with safranin O.

4. ST2 cells are an inappropriate cell to use for in vitro studies. Primary mesenchymal stem cells should be used to treat with the vesicles. No indication why the concentration of vesicles was used.

5. No indication as to why the time point of 7 days was chosen as the end point.

Minor Point

1. Graphs should show individual data points of each mouse in Fig 2.

6. PLOS authors have the option to publish the peer review history of their article (what does this mean?). If published, this will include your full peer review and any attached files.

Reviewer #1: No

Reviewer #2: **Yes: **Marley J. Dewey

Reviewer #3: No

---

## [Author Response · Author response to Decision Letter 0]

20 Dec 2022

Responses to the comments of Reviewer #1

Re: “Overall Consideration:”

“In the manuscript, “Matrix vesicles promote bone repair after a femoral bone defect in mice”, the authors utilize evaluate the effects of osteoblast-derived collagenase-released extracellular vesicles on cortical bone regeneration. The author utilized microCT and histology to assess bone healing within this model. This in vivo experiment in couple with in vitro analyses of osteoblast differentiation and mineralization to define mechanism. The authors argue that CREVs enhance healing within this model by increasing chondrogenesis that facilitates bone regeneration (e.g., endochondral ossification). This is an important topic of study, but there are some significant limitations. Overall, the depth and rigor of the study represent the largest weaknesses. A point-by-point review is listed below: “

(Response)

We would like to acknowledge the Reviewer’s helpful comments.

Re:” Major comment”

“1. Most of the study is observational, and the bit of mechanistic data presented by the authors are negative data.”

(Response)

We appreciate the Reviewer’s comment about the limitation of our study. We would like to clarify the detailed mechanisms by which osteoblast-derived CREVs enhance bone repair in the future study. The following comment was added as the limitation of the study in Discussion (page 30, line 473-475). “Most of our data were observational and lacking in mechanical insights. Therefore, further studies are necessary to clarify the mechanisms by which osteoblast-derived CREVs enhance bone repair.”

Re:” 2. The voxel size (25 microns) utilized in the microCT scanning is large and may be adding artifact to the authors' analyses. Moreover, this limits the ability of the authors to collect microarchitectural data that may point to mechanism(s).”

(Response)

Reviewer’s comments are helpful for our future study to clarify the mechanisms by which osteoblast-derived CREVs enhance bone repair. The following comment was added as the limitation of our study in Discussion (page 29, line 458-460), “µ-CT scanning with a voxel size of 25 µm may be low to analyze the microarchitectural data. µ-CT scanning with a higher resolution will provide insights into help the mechanisms by which osteoblast-derived CREVs enhance bone repair.”

Re:” 3. Did the authors euthanize mice prior to collection of the femora for analyses?”

(Response)

We collected femurs after fixation with 4% paraformaldehyde perfusion under anesthesia with 2% isoflurane. This comment was added in Materials and Methods (page 13, line 206-208).

Re:” 4. Can the authors demonstrate retainment of the applied scaffold applied within the defect following surgery?”

(Response)

Transplanted gelatin hydrogels were observed at the damaged site 7 days after the femoral bone defect in the present study. Moreover, we previously reported that myoblast derived EVs-containing gelatin hydrogels were degraded 9 days after transplantation at the site of the femoral bone defect (Takafuji et al., 2022, Endocr J.). These findings suggest that gelatin hydrogels are retained at damaged sites in femoral bone, and osteoblast-derived CREVs are released for 7 days. These comments were added to Discussion (page 25, line 393-page 26, line 396), and one reference [31] was added to References.

Reference

31. Takafuji Y, Kawao N, Ohira T, Mizukami Y, Okada K, Jo JI, et al. Extracellular vesicles secreted from mouse muscle cells improve delayed bone repair in diabetic mice. Endocr J. 2022 Oct 5. [Epub ahead of print].

Re:“5. Bone healing is a dynamic process. In the model, the authors are using it takes upwards of month for the defect to heal fully, so the day 7-time point the authors chose may not reflect the entire bone healing process. The authors need to discuss this and other limitations of the study.”

(Response)

The following comment was added to Discussion (page 29, line 454-page 29, line 457) as the limitation of our study, “We cannot completely rule out the possibility that osteoblast-derived CREVs exert different effects on the bone repair process at different time points after bone injury, because µ-CT and histological analyses were performed 7 days after the femoral bone defect.” 

Re:“6. The cortical bone defect model is supposed to be a model of intramembranous bone regeneration and we would not expect cartilage to form within this model. The authors observe some areas of metachromatic toluidine blue staining, and conclude that cartilage is forming within the defect. Toluidine blue can also stain osteoid in this manner. The authors should perform safranin O / fast green staining or some other more specific method to detect cartilage formation within the defect to substantiate further this claim.”

(Response) 

We added images of alcian blue-stained sections and the area of Alcian blue-stained region at the damaged sites 7 days after the femoral bone defect as new Fig 4B to Fig. 4 following the Reviewer’s comments. Alcian blue-staining data indicate that osteoblast-derived CREVs induce cartilage formation around the damaged sites in the femoral bone defect model we used. 

Cortical bone defect, such as the calvarial bone defect, is usually repaired through the intramembranous ossification, but not the enchondral ossification. However, the femoral bone defect we used is considered to be actually repaired through the intramembranous ossification and partly enchondral ossification, since the experimental procedure to make the pinhole bone defect without any damage of endosteum is practically difficult in our experiences for more than ten years. 

Re:”7. Standardized bone histomorphometry (e.g., osteoblasts per bone surface, osteoclasts per bone surface) would also help the authors assess the increased healing observed within their model. “

(Response) 

We measured the number of ALP-positive cells per bone surface at the damaged sites 7 days after the femoral bone defect following the Reviewer’s comments. The number of ALP-positive cells per bone surface was significantly increased at the bone damaged sites by local administration of CREVs 7 days after the femoral bone defect (Fig 3B). These comments were added to Results (page 20, lines 311-312), and new Fig 3B was added. 

Moreover, we measured the number of TRAP-positive multinucleated cells (MNCs) (osteoclastic cells) per bone surface at the damaged sites 7 days after the femoral bone defect was added as new Fig. 3D. The local administration of CREVs did not affect the number of TRAP-positive MNCs 7 days after the femoral bone defect (Fig 3D).This comment was added to Results (page 20, lines 314-page 21, line 316), and new Fig. 3D was added.

Re:”8. The periosteal reaction looks greater in the CREV treated group given the images provided. Can the authors quantify this? This may offer a clue to additional mechanism(s) for the authors. “

(Response)

Unfortunately, we could not quantify the periosteal reaction, because the identification of its border to the other sites was difficult for the quantitative analyses. The following comments were added to Discussion (page 28, line 443-page 29, line 448), “HE-stained images at the damaged site 7 days after the femoral bone defect showed that cells were clustered at the periosteal area of the damaged site after the femoral bone defect; however, we did not identify ALP, TRAP, or F4/80-positive cells in our preliminary immunohistochemical analyses (data not shown). We speculated that this periosteal reaction may be related to the mechanisms by which osteoblast-derived CREVs enhance bone repair in mice.”

Re:”9. The in vitro data supplied show that CREVs do not enhance osteoblast differentiation in vitro, so the authors conclude that enhanced chondrogenesis must be occurring. There are other explanations for their observations (e.g., altered osteoclastogenesis, increased progenitor recruitment from the periosteum, etc.). Direct data showing that CREVs enhance chondrogenesis and/or chondrocyte maturation (or some other mechanism) are needed to support claims made by the authors.”

(Response)

Although we concluded that the local administration of osteoblast-derived CREVs with a gelatin hydrogel enhance bone repair partly through osteogenesis and chondrogenesis at the damaged sites after femoral bone injury in mice, the mechanism by which osteoblast-derived CREVs enhances osteogenesis and chondrogenesis have still remained unknown in our study, as we discussed. Following the Reviewer’s comments, the following sentence was added to Discussion as follows (page 29, line 449-451), “Alternatively, the regulation of inflammation, the alternation of osteoclastogenesis, increased progenitor recruitment from the periosteum, or direct effects on chondrogenesis may be related to these mechanisms.”

Re:” Minor Comments:”

“1. The grammar issues within the manuscript need to be addressed.”

(Response)

We asked the native English speaker for the edition of English in the manuscript.

Responses to the comments of Reviewer #2: 

Re:“The authors present an interesting article characterizing collagenase-digested EVs and the use of these in a femoral defect model and the impact of these on osteogenesis in vitro. Some claims of the EV identification are debatable, some methodology is missing/unclear, and in vitro assessments should more closely match in vivo application of CREVs in a hydrogel. The following major revisions are requested: “

(Response)

We acknowledge the Reviewer’s kind comments.

Re:” Introduction:”

“1. The introduction is lacking detail about the specific bone defect the authors are addressing. The authors claim that a bone defect is generally repaired without clinical intervention and later mention that clinical intervention is needed in some instances of critical-sized defect repair. Authors should re-structure their introduction to bone repair with the focus of femoral defect repair, such as current methods and needs for new approaches, and limitations of current approaches for this specific repair problem. “

(Response)

The first sentence of Introduction was corrected as the follow in Introduction (page 3, liens 35-36), “Bone tissues possess high regeneration capacity, and minor bone defect is generally repaired without any clinical intervention [1, 2].”

 A femoral bone defect model has generally been used to investigate the therapeutic effects of substitute biomaterials on bone repair and the bone repair process in mice as a representative model of the bone repair process after bone defects in various bones as well as a femoral bone defect (Li Y et al., 2015, J Orthop Translat; Inzana JA et al. 2014, Biomaterials). On the other hand, a calvarial bone defect model is sometimes useful for assessing bone repair through membranous ossification. Since the repair of the calvaria after a bone defect is difficult without the addition of strong osteogenic factors, such as bone morphogenetic proteins, we selected the femoral bone defect model to investigate the effects of osteoblast-derived CREVs on bone repair in mice. These sentences were added to Introduction (page 5, line 81-page 6, line 89), and two references [27, 28] were added to References.

Reference 

27. Li Y, Chen SK, Li L, Qin L, Wang XL, Lai YX. Bone defect animal models for testing efficacy of bone substitute biomaterials. J Orthop Translat. 2015 Jun 16;3(3):95-104.

28. Inzana JA, Olvera D, Fuller SM, Kelly JP, Graeve OA, Schwarz EM et al. 3D printing of composite calcium phosphate and collagen scaffolds for bone regeneration. Biomaterials. 2014 Apr;35(13):4026-34.

Re:” Methods:”

“2. Collagenase usage is listed as Units/mL. Define these arbitrary ‘units’. “

(Response)

The following comment were added in Materials and Methods (page 7, line 102-,103), “One collagenase I unit liberated 1 µmol of L-leucine equivalents from collagen in 5 h at 37°C, pH 7.5.” 

Re:” 3. ALP activity is missing methods. Based on figures, it seems this assay was performed on the CREV. Add details explaining what samples were used and how much of these samples were used in this assay.”

(Response)

The following sentences were added to Material and Methods (page 11, line 168-176), “Collagenase-treated mouse osteoblasts and CREVs were lysed with RIPA buffer after the CREV isolation process. Lysed cells and CREVs were centrifuged at 12,000×g at 4°C for 10 min. Supernatants were collected as protein samples. Mouse osteoblasts were cultured with or without 4 µg/mL CREVs for 72 h, cells in 24-well plates were rinsed three times with PBS, and 200 µL of distilled water was then added to each well. Lysed cell homogenates were centrifuged at 12,000×g at 4°C for 10 min. Supernatants were collected as protein samples. After protein levels were quantified using BCA assay reagent, protein samples were diluted to 10 or 1 µg protein/mL and analyzed with LabAssay ALP (FUJIFILM Wako Chemicals) according to the manufacturer’s protocol.”

Re: “4. Authors utilize a gelatin hydrogel to deliver CREV to the bone defect site; however, there is no methodology on these hydrogels. Authors need to include information on methods of gelatin hydrogel production, characteristics of hydrogels (porosity, mechanics, etc.), and release characteristics of encapsulated EVs. If this is already published work, then this should be added to the introduction explaining these findings.”

 (Response)

The following sentence was added to Introduction (page 5, lines 80-81), “Cationized gelatin hydrogels are suitable for the local sustained release of negatively charged molecules due to degradation [26].”

The following sentence was added to Discussion (page 25, line 389-page 26, line 396), “Cationized gelatin hydrogels are negatively charged molecules due to degradation, and EVs possess a negative charge on their surface; therefore, cationized gelatin hydrogels are used as their carrier for local administration [31, 39]. Transplanted gelatin hydrogels were observed at the damaged site 7 days after the femoral bone defect in the present study. Moreover, we previously reported that myoblast-derived EV-containing gelatin hydrogels were degraded 9 days after transplantation at the site of the femoral bone defect [31]. These findings suggest that gelatin hydrogels are retained at damaged sites in femoral bone, and osteoblast-derived CREVs are released for 7 days.”

Two references [26, 39] were added to References.

Reference

26. Mishima S, Takahashi K, Kiso H, Murashima-Suginami A, Tokita Y, Jo JI, et al. Local application of Usag-1 siRNA can promote tooth regeneration in Runx2-deficient mice. Sci Rep. 2021 Jul 1;11(1):13674.

39. Ikebuchi Y, Aoki S, Honma M, Hayashi M, Sugamori Y, Khan M, et al. Coupling of bone resorption and formation by RANKL reverse signaling. Nature. 2018 Sep;561(7722):195-200.

Re: “5. What was the dosing of CREV used in the mineralization assay? Add this to the methods section.”

(Response)

4 µg/mL osteoblast-derive CREVs were used in the mineralization assay and the information was added to Materials and Methods (page 16, line 243-244).

Re: “6. Cell isolation- how were osteoblast cells verified for correct cell isolation? Information on cell line manufacturer for ST2 cells is also needed.”

(Response)

The information about primary osteoblasts was added to Materials and Methods as the follow (page 8, line 123-124), “Isolated osteoblasts exhibited ALP activity and formed mineralized nodules.” 

ST-2 cells were obtained from RIKEN, Tsukuba, Japan, and the information was added to Materials and Methods (page 7, line 115).

Re: “Discussion”

“7. Authors seem to be claiming that CREVs may be secreted in an exosome-like fashion, but in order to claim this, authors must provide more supporting information than CD9+ and similar size range. This sentence could be removed instead.”

(Response)

The followed sentence was removed from Discussion in the original manuscript (page 23, line 347-348, in the original manuscript) “These findings suggest that exosome-like EVs in CREVs might be similar with the MtVs.”

Re: “8. Authors use CREVs and MtEVs interchangeably but note that CREVs may contain exosome-like EVs and MtEVs, but later claim MtEVs induce bone repair based on CREV results. Authors should clarify the identification of their EVs (likely MtEVs) and use this consistently within the manuscript. For clarity, authors should consider only using MtEVs as an acronym for their EV used in this study.”

 (Response)

“Matrix vesicles (MtVs)” has been defined and used as vesicles existing in the extracellular matrix based on the histological analysis with the physiological function as an initiator of mineralization. MtVs are considered to be included in the subtypes of extracellular vesicles (EVs). Electron micrograph study of bone tissue showed that MtVs are extracellular vesicles with diameter of 100-300 nm, and their secretion are thought to be a microvesicle-like mechanism, budding from the plasma membrane [17]. As we discussed, EVs are classified into three main subtypes of vesicles based on the vesicle formation mechanism; exosomes, microvesicles, and apoptotic bodies [7, 8]. Exosomes with 50-150 nm in diameter are formed by endosomal route [29]. Microvesicles with 50-1000 nm in diameter are formed by budding from the plasma membrane [30]. Apoptotic bodies with 500-2000 nm in diameter are formed by plasma membrane blebbing [31]. Although the differences between MtVs and exosomes or microvesicles still remained unclear at the present time, the specific character of MtVs are thought to be high affinity with extracellular matrix, compared to most EVs. Taken together, we cannot rule out the possibility that collagenase released extracellular vesicles (CREVs) are not identical to MtVs concisely. Moreover, the position statement of ISEV2018 claimed that “unless authors can establish specific markers of subcellular origin that are reliable within their experimental system(s), authors are urged to consider use of operational terms for EV subtypes. We therefore used the word, “collagenase released extracellular vesicles (CREVs)” for the experimental description in the present study. On the other hand, we used MtVs for the general statement. Although we believe that the different use of CREVs and MtVs are scientifically neutral, they are actually the same in our manuscript. 

Re: “9. Authors assess osteogenesis in a 2D model in vitro but to more accurately represent in vivo work, authors should examine the response of osteogenic cells seeded on hydrogels containing CREVs.”

(Response)

The following comments were added to Discussion as follows (page 30, line 469-475) as the limitation of our study, “Another limitation is that in the in vivo study, we used a gelatin hydrogel as the osteoblast-derived CREV carrier and cellular scaffold. Therefore, a 2-dimensional culture system may be insufficient to estimate the response of osteogenic cells to osteoblast-derived CREVs.”

Re: “10. There was a lack of mineralization differences in vitro and authors suggest CREVs could work through a different mechanism. Authors should consider examining cell activity in response to CREVs and protein release of those related to osteoclastogenesis and inflammation (OPG, IL6, etc.), especially from mesenchymal stem cells, as this may prove interesting mechanistic results of CREVs.”

(Response)

We appreciate the Reviewer’s helpful comments. Since we could not show the data indicating the detailed mechanical insights, Discussion was corrected as the follows.

Discussion (page 28, line 437-page 29, line 453)

“Bone repair proceeds in three phases: inflammation, renewal, and remodeling. The cells recruited to the damaged site in bone in the inflammation phase induce the migration of osteogenic cells. For example, macrophages are recruited and induce the differentiation of mesenchymal stem cells into osteoblasts by secreting transforming growth factor-β, insulin-like growth factor-1, and fibroblast growth factor 2 [41]. Therefore, MtVs may promote bone repair through cells other than osteoblasts or mesenchymal cells. HE-stained images at the damaged site 7 days after the femoral bone defect showed that cells were clustered at the periosteal area of the damaged site after the femoral bone defect; however, we did not identify ALP, TRAP, or F4/80-positive cells in our preliminary immunohistochemical analyses (data not shown). We speculated that this periosteal reaction may be related to the mechanisms by which osteoblast-derived CREVs enhance bone repair in mice. Alternatively, the regulation of inflammation, the alternation of osteoclastogenesis, increased progenitor recruitment from the periosteum, or direct effects on chondrogenesis may be related to these mechanisms. Otherwise, MtVs may enhance bone repair through the supply of hydroxyapatite for mineralization. Further studies are needed to clarify this issue.”

Discussion (page 30, line 472-475),

“In addition, most of our data were observational and lacking in mechanical insights. Therefore, further studies are necessary to clarify the mechanisms by which osteoblast-derived CREVs enhance bone repair.”

Responses to the comments of Reviewer #3

Re: “This study by Mizakumi et al. isolates vesicles from mouse osteoblast cultures. Vesicles were used to treat a femoral defect and osteoblast cultures. There are several major concerns with experimental design and methods of the study that make it difficult to interpret the results.”

(Response)

We acknowledge the Reviewer’s helpful comments.

Re: “1. There is no indication of the sex or age of the mice used in the femoral defect model experiment. No power analysis is included to indicate how number of animals used was determined.”

(Response)

Twelve-week-old female C57BL/6 mice were used for the femoral defect model in the present study. This information was added to Materials and Methods (page 11, line 179-180). 

The sample size for each experiment was selected using G*Power 3.1 software (https://gpower.software.informer.com/3.1/). We used parameters based on our previous studies for a power analysis. These comments were added to Materials and Methods (page 17, line 251-253). 

Re: “2. ALP cells in Fig 3 should be quantitated per number of DAPI number of cells.”

(Response)

We measured the number of ALP-positive cells per number of DAPI-positive nucleus at the bone damaged sites 7 days after the femoral bone defect following the Reviewer’s comments. The number of ALP-positive cells per number of DAPI-positive nucleus was significantly increased at the bone damaged sites by local administration of CREVs 7 days after the femoral bone defect (Fig 3C). These comments were added to Results (page 20, lines 311-312), and new Fig 3C was added. 

Re: “3. Sections stained with toluidine blue to indicate areas of cartilage should instead be stained with safranin O.”

(Response)

We apologized not to perform safranin O staining. We instead added images of Alcian blue-stained sections at the damaged sites 7 days after a femoral bone defect as new Fig. 4B. Alcian blue-staining data indicated that osteoblast-derived CREVs induce cartilage formation around the damaged sites in the femoral bone defect model we used. 

Re: “4. ST2 cells are an inappropriate cell to use for in vitro studies. Primary mesenchymal stem cells should be used to treat with the vesicles. No indication why the concentration of vesicles was used.”

(Response)

The following comments were added to Discussion (page 29, line 460-page 30, line 469) as the limitation of the study, “Moreover, in the in vitro study, the effects of osteoblast-derived CREVs on osteoblastic differentiation were estimated using ST2 cells. Experiments using primary mesenchymal stem cells may be necessary to clarify the effects of osteoblast-derived CREVs on the differentiation of mesenchymal stem cells into osteoblasts in vitro because ST2 cells may not represent mesenchymal stem cells. A previous study reported that osteoblast-derived EVs exhibited biological activity at concentrations of 1-2.5 µg/mL [16, 24]. Although osteoblast-derived CREVs were used at a concentration of 4 µg/mL in the in vitro study without significant osteogenic effects, the concentration of osteoblast-derived CREVs may have been too low to exert significant osteogenic effects.”

Re: “5. No indication as to why the time point of 7 days was chosen as the end point.”

(Response)

The following comment was added to Materals and Methods (page 12, line 190-193), “New bone started to form, and ALP-positive cells were observed at the damaged area 7 days after the femoral bone defect, as previously reported [31]. Therefore, we selected the time point of 7 days after the bone defect to assess bone repair in the present study.”, and new reference [31] was added to References.

References

31. Takafuji Y, Kawao N, Ohira T, Mizukami Y, Okada K, Jo JI, et al. Extracellular vesicles secreted from mouse muscle cells improve delayed bone repair in diabetic mice. Endocr J. 2022 Oct 5. [Epub ahead of print].

Re: “Minor Point”

“1. Graphs should show individual data points of each mouse in Fig 2.”

(Response)

Individual data points of each mouse were added in Fig 2.

---

## [Decision Letter · Decision Letter 1]

27 Jan 2023

PONE-D-22-31340R1Matrix vesicles promote bone repair after a femoral bone defect in mice.PLOS ONE

Dear Dr. Kaji,

Thank you for submitting your manuscript to PLOS ONE. After careful consideration, we feel that it has merit but does not fully meet PLOS ONE’s publication criteria as it currently stands. Therefore, we invite you to submit a revised version of the manuscript that addresses the points raised during the review process.

We look forward to receiving your revised manuscript.

Kind regards,

Isha Mutreja

Academic Editor

PLOS ONE

Reviewers' comments:

Reviewer's Responses to Questions

**Comments to the Author**

1. If the authors have adequately addressed your comments raised in a previous round of review and you feel that this manuscript is now acceptable for publication, you may indicate that here to bypass the “Comments to the Author” section, enter your conflict of interest statement in the “Confidential to Editor” section, and submit your "Accept" recommendation.

Reviewer #1: (No Response)

Reviewer #2: (No Response)

Reviewer #3: (No Response)

2. Is the manuscript technically sound, and do the data support the conclusions?

Reviewer #1: No

Reviewer #2: Yes

Reviewer #3: Yes

3. Has the statistical analysis been performed appropriately and rigorously? 

Reviewer #1: Yes

Reviewer #2: Yes

Reviewer #3: Yes

4. Have the authors made all data underlying the findings in their manuscript fully available?

Reviewer #1: Yes

Reviewer #2: No

Reviewer #3: Yes

5. Is the manuscript presented in an intelligible fashion and written in standard English?

Reviewer #1: Yes

Reviewer #2: Yes

Reviewer #3: Yes

6. Review Comments to the Author

Reviewer #1: The authors address some of the comments within their revised manuscript, but there is still a disconnect between the in vivo observations and the in vitro work done to explain those data. There periosteal reaction with the CREV treatment in enormously different, but that is not explored whatsoever.

Reviewer #2: The original comment about the gelatin hydrogels was not fully addressed. Authors have partially answered this comment, but must add some greater detail to the fabrication process of these hydrogels in the methods section to ensure reproducibility. Briefly list fabrication process including how CREVs were encapsulated within the hydrogel into the methods section with references on prior use of these hydrogels.

Reviewer #3: The authors have addressed all my comments to my satisfaction. However, if they can address the comments made by other reviewers, it will improve the scientific merit of the manuscript.

7. PLOS authors have the option to publish the peer review history of their article (what does this mean?). If published, this will include your full peer review and any attached files.

Reviewer #1: No

Reviewer #2: **Yes: **Marley Dewey

Reviewer #3: No

---

## [Author Response · Author response to Decision Letter 1]

12 Mar 2023

Responses to the comments of Reviewer #1

Re: “The authors address some of the comments within their revised manuscript, but there is still a disconnect between the in vivo observations and the in vitro work done to explain those data. There periosteal reaction with the CREV treatment in enormously different, but that is not explored whatsoever.”

(Response)

We appreciated to the Reviewer’s important remarks on our study. We performed some additional experiments to evaluate the periosteal reaction with the CREV treatment and clarify the mechanisms by which CREVs transplantation enhances chondrogenesis, osteoblastogenesis and bone repair after the femoral bone defect in mice in contrast to the in vitro data. Unfortunately, we could not show the vehicle group for Fig. S2 for the lack of residual tissue sections for immunohistochemistry, although they were available for only the evaluation of PDGFRα- and SDF-1-positive cells, which were most important for the additional experiments. Area of larger number of cell accumulation was observed at periosteum site 7 days after the femoral bone defect with CREVs transplantation, compared to vehicle group (Fig. 2, S1). Numerous PDGFRα-positive cells (skeletal stem/progenitor cells (SSPCs)) and Osterix-positive cells (preosteoblastic cells) were observed at periosteum area 7 days after the femoral bone defect with CREVs transplantation (Fig. S2, S3), although few ALP-positive cells (osteoblastic cells), F4/80-positive cells (macrophages), TRAP-positive cells (osteoclasts) and CD3-positive cells (T-lymphocytes) were detected (Fig. S2). In addition, the number of SDF1-positive cells seemed to be increased at the periosteum area by the local administration of CREVs 7 days after the femoral bone defect (Fig S4). These sentences were added to Results (page 23, lines 364-374), and new supplemental Figures (Figure S1-S4) were added. 

 The following Discussion (page 30, lines 479-495) and 8 references [44-51] were added to assess some mechanisms by which CREVs transplantation enhances chondrogenesis, osteoblastogenesis and bone repair after the femoral bone defect in mice in contrast to the in vitro data, “SSPCs are cells with the characteristics of mesenchymal stem cells as well as mobilized from bone marrow, periosteum and skeletal muscles during bone regeneration [44]. SSPCs possess different regenerative potential according to their origin. Namely, SSPCs from bone marrow, periosteum and muscles contribute to osteoblastic bone formation, osteochondral bone formation and cartilage formation, respectively [45-47]. Since the local administration of CREVs seemed to increase numbers of PDGFRα- and Osterix-positive cells as well as cartilage/bone formation at the bone damaged sites in our study, CREVs might recruit SSPCs from bone marrow, periosteum and skeletal muscles, then improving bone regeneration in vivo. Our data showed that numerous SDF1-positive cells were observed at the periosteal area of the damaged site with the local administration of CREVs. SDF1 is a chemokine involved in the recruitment of mesenchymal stem cells at the fracture sites [48-51]. Taken together, the local administration of CREVs might recruit PDGFα-positive SSPCs at the bone damaged sites partly through SDF1, leading to an enhancement of chondrogenic/osteoblastic bone formation and subsequent bone repair in mice. Alternatively, MtVs may enhance bone repair through the supply of hydroxyapatite for mineralization. Further studies are needed to clarify this issue.”

Reference

44. Julien A, Perrin S, Martínez-Sarrà E, Kanagalingam A, Carvalho C, Luka M, Ménager M, Colnot C, Skeletal Stem/Progenitor Cells in Periosteum and Skeletal Muscle Share a Common Molecular Response to Bone Injury. J Bone Miner Res. 2022 Aug;37(8):1545-1561.

45. Méndez-Ferrer S, Michurina TV, Ferraro F, Mazloom AR, Macarthur BD, Lira SA, Scadden DT, Ma'ayan A, Enikolopov GN, Frenette PS, Mesenchymal and haematopoietic stem cells form a unique bone marrow niche. Nature. 2010 Aug 12;466(7308):829-34.

46. Duchamp de Lageneste O, Julien A, Abou-Khalil R, Frangi G, Carvalho C, Cagnard N, Cordier C, Conway SJ, Colnot C, Periosteum contains skeletal stem cells with high bone regenerative potential controlled by Periostin. Nat Commun. 2018 Feb 22;9(1):773.

47. Julien A, Kanagalingam A, Martínez-Sarrà E, Megret J, Luka M, Ménager M, Relaix F, Colnot C, Direct contribution of skeletal muscle mesenchymal progenitors to bone repair. Nat Commun. 2021 May 17;12(1):2860.

48. Herrmann M, Verrier S, Alini M, Strategies to Stimulate Mobilization and Homing of Endogenous Stem and Progenitor Cells for Bone Tissue Repair. Front Bioeng Biotechnol. 2015 Jun 2;3:79.

49. Link DC, Neutrophil homeostasis: a new role for stromal cell-derived factor-1. Immunol Res. 2005;32(1-3):169-78.

50. Okada K, Kawao N, Yano M, Tamura Y, Kurashimo S, Okumoto K, Kojima K, Kaji H, Stromal cell-derived factor-1 mediates changes of bone marrow stem cells during the bone repair process. Am J Physiol Endocrinol Metab. 2016 Jan 1;310(1):E15-23.

51. Dar A, Kollet O, Lapidot T, Mutual, reciprocal SDF-1/CXCR4 interactions between hematopoietic and bone marrow stromal cells regulate human stem cell migration and development in NOD/SCID chimeric mice. Exp Hematol. 2006 Aug;34(8):967-75.

Responses to the comments of Reviewer #2

Re: “The original comment about the gelatin hydrogels was not fully addressed. Authors have partially answered this comment, but must add some greater detail to the fabrication process of these hydrogels in the methods section to ensure reproducibility. Briefly list fabrication process including how CREVs were encapsulated within the hydrogel into the methods section with references on prior use of these hydrogels.”

(Response)

We apologized for the insufficient explanation about the fabrication process of cationized gelatin hydrogels. The paragraph about “Preparation of cationized gelatin hydrogel sheets” in Materials and Methods was corrected to the follow (page 11, lines 179-197), “Cationized gelatins were synthesized by converting the carboxyl groups of gelatin with the amino groups, previously described [30]. Briefly, 7.8 g ethylenediamine (EDA) was added into 250 ml of gelatin solution (25 mg/ml) in 0.1 M phosphate-buffered solution (PB, pH = 5.0) at 40 °C. The pH of the solution was adjusted to 5.0 by adding 11 M HCl aqueous solution. 5.35 g ethylenedichloride (EDC) was added into the solution and PB was added into the solution to give the final volume of 500 ml. After stirring at 40 °C for 18 h, the gelatin solution was dialyzed against double-distilled water (DDW) for 3 days at room temperature. The dialyzed solution was freeze-dried to obtain cationized gelatins. To determine the percentage of amino groups introduced into gelatin, the conventional 2,4,6-trinitrobenzene sulfonic acid method was performed [31]. The percentage was 19.9 mole% per the carboxyl groups of gelatins. 1.2 ml of cationized gelatin aqueous solution (5.0 mg/ml) was poured into polytetrafluoroethylene mold (43 mm × 43 mm) and freeze-dried to prepare the cationized gelatin hydrogel sheets. The hydrogel sheets were stabilized by dehydrothermal crosslinking at 160℃ for 72 h with dry oven (AURORA DN-305, Sato Vacuum Inc., Japan). Gelatin hydrogel sheets were punched out to circular disc with the diameter of 1.5 mm for the transplantation. 1.5 µL of CREVs suspension in PBS (6.67 µg/µL) was soaked into disc-shaped gelatin hydrogel.”, and two references [30, 31] were added to References.

Reference

30. Murata Y, Jo JI., Tabata Y, Preparation of cationized gelatin nanospheres incorporating molecular beacon to visualize cell apoptosis. Sci Rep. 2018 Oct 4;8(1):14839.

31. Snyder SL, Sobocinski PZ, An improved 2,4,6-trinitrobenzenesulfonic acid method for the determination of amines. 1975 Mar;64(1):284-8.

 

Responses to the comments of Reviewer #3

Re: The authors have addressed all my comments to my satisfaction. However, if they can address the comments made by other reviewers, it will improve the scientific merit of the manuscript.

(Response)

We appreciated for the Reviewer’s positive comments for our manuscript. We added some new data and revised our manuscript to address the other Reviewer's comments.

---

## [Decision Letter · Decision Letter 2]

27 Mar 2023

Matrix vesicles promote bone repair after a femoral bone defect in mice.

PONE-D-22-31340R2

Dear Dr. Kaji,

We’re pleased to inform you that your manuscript has been judged scientifically suitable for publication and will be formally accepted for publication once it meets all outstanding technical requirements.

Kind regards,

Isha Mutreja

Academic Editor

PLOS ONE

Additional Editor Comments (optional):

Reviewers' comments:

Reviewer's Responses to Questions

**Comments to the Author**

1. If the authors have adequately addressed your comments raised in a previous round of review and you feel that this manuscript is now acceptable for publication, you may indicate that here to bypass the “Comments to the Author” section, enter your conflict of interest statement in the “Confidential to Editor” section, and submit your "Accept" recommendation.

Reviewer #1: All comments have been addressed

2. Is the manuscript technically sound, and do the data support the conclusions?

Reviewer #1: Yes

3. Has the statistical analysis been performed appropriately and rigorously? 

Reviewer #1: Yes

4. Have the authors made all data underlying the findings in their manuscript fully available?

Reviewer #1: Yes

5. Is the manuscript presented in an intelligible fashion and written in standard English?

Reviewer #1: Yes

6. Review Comments to the Author

Reviewer #1: The authors have revised this manuscript sufficiently. Thank you for the additional edits to this manuscript.

7. PLOS authors have the option to publish the peer review history of their article (what does this mean?). If published, this will include your full peer review and any attached files.

Reviewer #1: No

---

## [Editor Report · Acceptance letter]

31 Mar 2023

PONE-D-22-31340R2 

Matrix vesicles promote bone repair after a femoral bone defect in mice. 

Dear Dr. Kaji:

I'm pleased to inform you that your manuscript has been deemed suitable for publication in PLOS ONE. Congratulations! Your manuscript is now with our production department. 

Kind regards, 

on behalf of

Dr. Isha Mutreja 

Academic Editor

PLOS ONE